# Native and Non-Indigenous Biota Associated with the *Cymodocea nodosa* (Tracheophyta, Alismatales) Meadow in the Seas of Taranto (Southern Italy, Mediterranean Sea)

**Giuseppe Denti, Fernando Rubino, Ester Cecere and Antonella Petrocelli \***

Institute for Water Research (IRSA)-CNR, via Roma 3, 74123 Taranto, Italy; giuseppe.denti@cnr.it (G.D.);
fernando.rubino@cnr.it (F.R.); ester.cecere@cnr.it (E.C.)

\* Correspondence: antonella.petrocelli@cnr.it

**Abstract:** The collection of photos during the systematic monitoring activities is useful to witness the ecological role of marine phanerogams as hosts for a rich variety of organisms in coastal and transitional waters. *Cymodocea nodosa* is present in the Taranto seas. In Mar Piccolo, it reached high coverage in a short amount of time, up to 100%, due to the improvement in environmental conditions. The most recent observations showed that it offers a welcoming habitat for several vertebrates and invertebrates, native and non-indigenous, as well as to micro- and macroalgae. The NPPR-funded activities will make these observations more robust and structural.

**Keywords:** biodiversity; ecosystem services; ITINERIS; LTER; Mar Piccolo of Taranto; non-indigenous species; NBFC; seagrasses

Marine phanerogams are paramount for the structure and function of marine coastal ecosystems. They are emblematic primary producers, protecting shorelines from erosion, supporting high biodiversity, and offering food, refuge, and nursery areas to invertebrates and vertebrates, such as fish and even migrant birds [1–4]. Their value in terms of ecosystem services, in the Caribbean alone, was estimated at about USD 255 billion per year [5]. The abundance of marine phanerogams is a sign of excellent environmental conditions; thus, they have been included among the primary species for the definition of priority habitat within the Habitat Directive (92/43/EEC) and used as bioindicators in the formulation of biotic indices for the assessment of good environmental status (GES) in coastal and transitional water systems, according to the Water Framework Directive (2000/60/EC) and the Marine Strategy Framework Directive (2008/56/EC) [6–8].

*Cymodocea nodosa* (Ucria) Ascherson is a plastic marine phanerogam, able to flourish in pristine environments and survive in stressed ones [9,10]. It is widely distributed in the Mediterranean coastal and transitional water systems, from east to west [9,11]. The articulated rhizome–root component, with dense slender leaves, forms a robust grid, which stabilizes the substrate and welcomes the vagile fauna. Grazers also find food in the rich epiphytic community and often spawn among the shoots [2,3,12]. A value of about EUR 3 million per year was recently assessed for the commercial fish community housed by the *C. nodosa* meadows of the Canary Islands [13]. Seahorses are joint guests of this marine plant [14,15], thus it has earned the folk name of "seahorse grass" [16].

*Cymodocea nodosa* is present with a dense meadow along the south-eastern coast of the Mar Grande of Taranto [F. Rubino personal communication] (Figure 1), where it sexually reproduces [E. Cecere personal communication]. Underwater observations by scuba divers showed that it hosts varied fauna from native seahorses, such as *Hyppocampus hippocampus* (Linnaeus, 1758) and *H. guttulatus* (Cuvier 1829), to non-indigenous mollusks, such as *Pinctada radiata* (Leach, 1814) (Figure 2a,b).

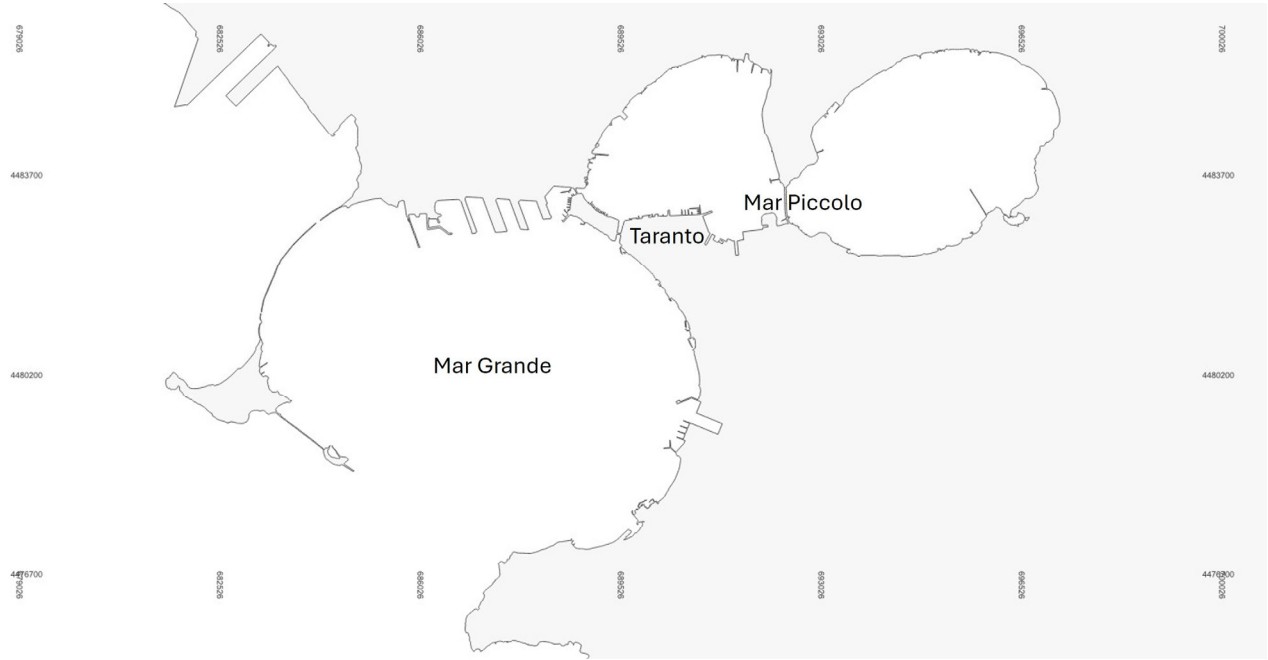

**Figure 1.** Taranto is known as the city of two seas. A map of the marine area with Mar Grande and Mar Piccolo. The numbers indicate UTM coordinates.

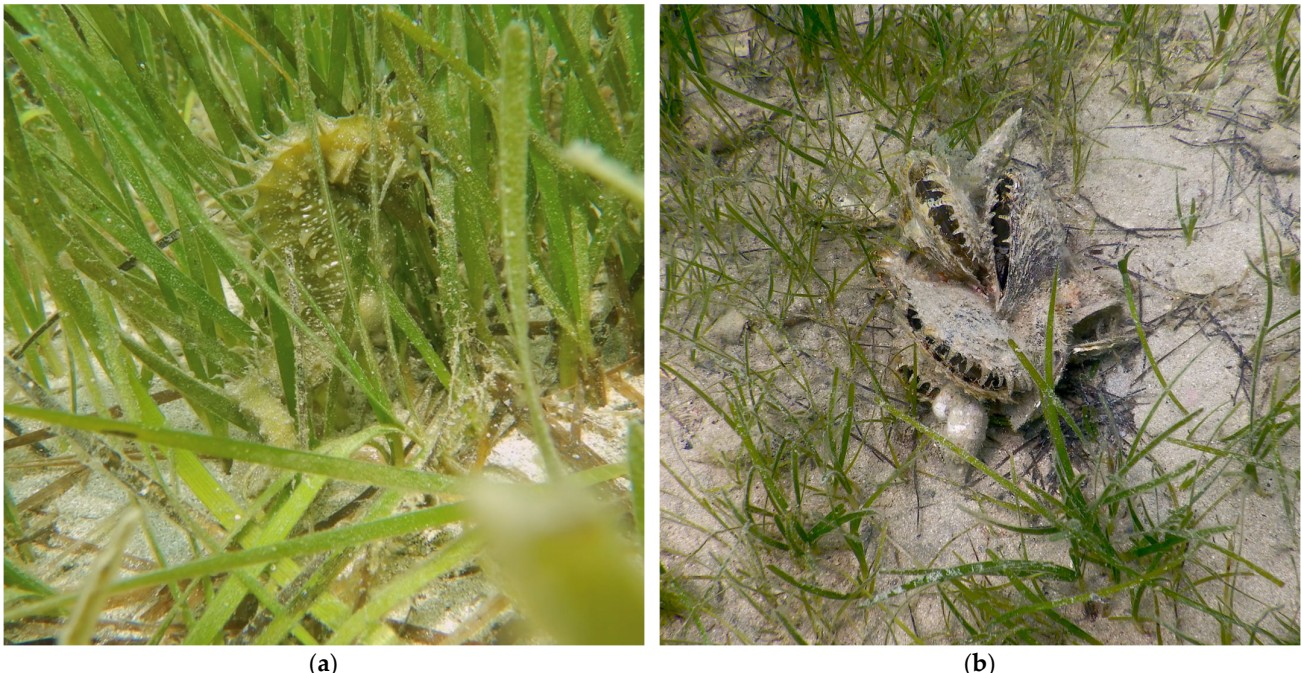

(**a**)    (**b**)

**Figure 2.** In the Mar Grande of Taranto, a *Cymodocea nodosa* meadow flourishes along the southeastern coast, and hosts a diverse associated fauna among its shoots: (**a**) the seahorse *Hyppocampus guttulatus* takes refuge within the meadow; (**b**) aggregated specimens of *Pinctada radiata*.

In Mar Piccolo, a transitional water system located in the Ionian Sea (Figure 1), after a consistent thinning in the 1980s and the 1990s, due to the presence of high eutrophication, *C. nodosa* is now flourishing again [17]. It is widely distributed along the basin coast, and in some zones, the meadow reaches a cover of up to 100% [18], a sign of a high ecological status, according to the MaQI index [6] (Figure 3).

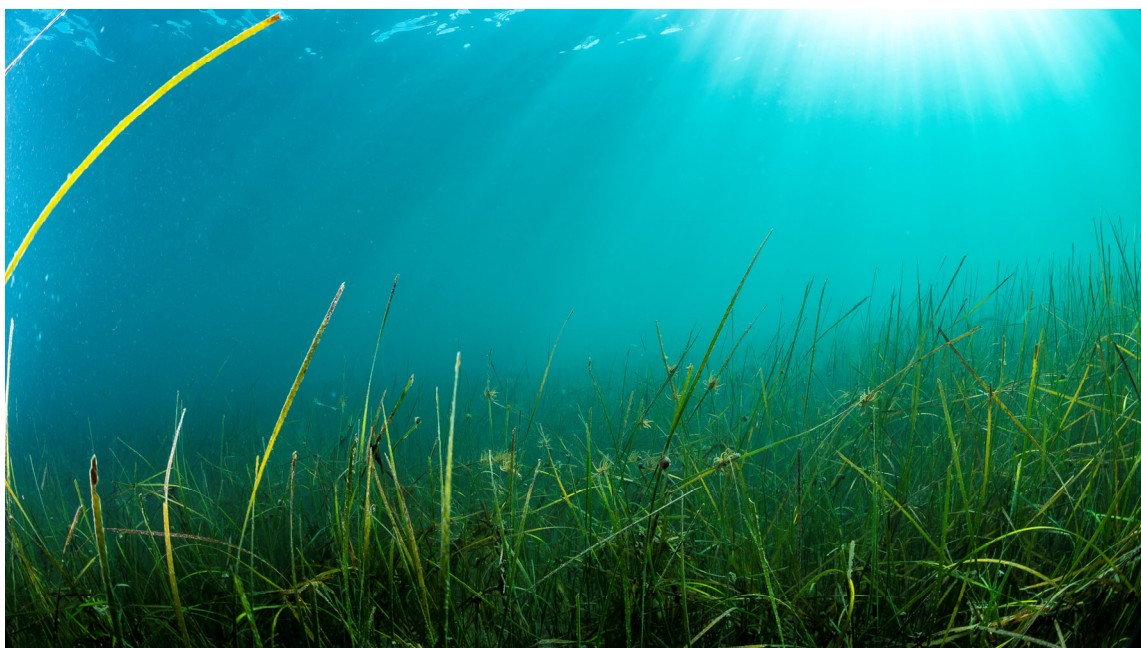

**Figure 3.** Starting from 2000, due to improving environmental conditions, a dense meadow of *Cymodocea nodosa* developed on the bottom of the Mar Piccolo basin (Photo credits: Gianni Squitieri).

The seasonal systematic observations carried out on the Mar Piccolo macrobenthos, mainly phytobenthos, since 2011, within the framework of long-term ecological research (LTER) activities, currently funded by the National Recovery and Resilience Plan (NRRP) Project ITINERIS, has allowed a visual assessment of the progressive recovery of the meadow [17]. In addition, the activities performed within the framework of the Project "MIA Natura 2000", funded through POR PUGLIA FESR-FSE 2014/2020, led to the quantitative evaluation of the meadow's density, as well as its structural descriptors [18]. Phenological monitoring is now continued within the NRRP, with the Project "National Biodiversity Future Center". The underwater observations carried out in these years, for the collection of benthos samples, also brought to light the close relationships of *C. nodosa* with the fauna and flora living in the basin.

For several years, starting from 2014, it was common to see specimens of the fan mussel *Pinna nobilis* (Linnaeus 1758) among the shoots of *C. nodosa* [19] (Figure 4). The big bivalve mollusk shows a marked preference for this phanerogam in transitional waters, mainly due to the close bonds produced among the rhizomes, the sediments, and the byssus filaments [20–23]. Also of paramount importance is the protective effect of leaves against hydrodynamics, which fosters the mollusk's larvae settlements [23]. In Spain, in the transitional system of the Ebro River (NW Mediterranean), the highest density of *P. nobilis* specimens were measured in zones where the *C. nodosa* meadow reached a cover of 80–100% [21]. In the Venice Lagoon (Adriatic Sea, northern Italy), the fan mussel density was the highest at the highest cover of the phanerogam, and it was preferably located at the border of the meadow, where the filtering activity can be enhanced [23]. Unfortunately, Figure 4 represents a historical relic; starting in 2018, *P. nobilis* has undergone a mass mortality in the Taranto seas, due to an infection by the protozoic parasite, *Haplosporidium pinnae* [24]. This caused its disappearance from the basin, as observed all around the Mediterranean Sea [25].

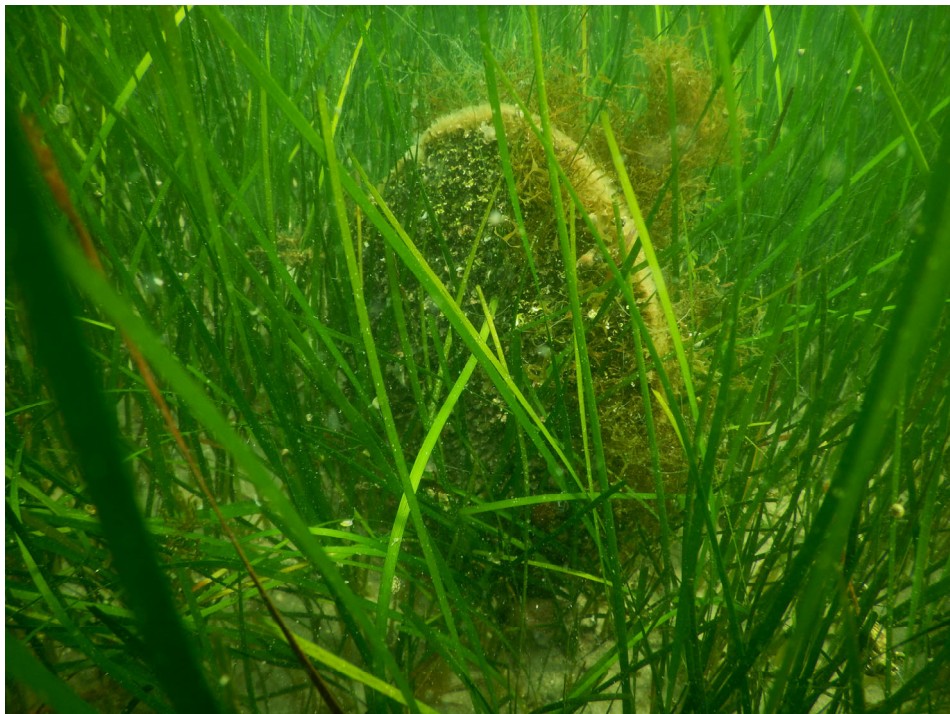

**Figure 4.** Until 2018, it was common to observe healthy specimens of *Pinna nobilis* among the leaves of *Cymodocea nodosa* in Mar Piccolo of Taranto (Photo credits: Gianni Squitieri).

Mar Piccolo of Taranto, like most transitional water systems, is subject to a high degree of biological pollution. At the moment, 22 non-indigenous invertebrates and 16 seaweeds have been recorded in the basin [26], A. Petrocelli (unpublished data). Generally, marine non-indigenous species (NIS) are considered a serious menace for native communities, especially when they have a markedly invasive behavior, since they can cause strong alteration in ecosystem functions, reduction in biodiversity, and damages to economic activities [27,28]. *Caulerpa cylindracea* Sonder considerably spread along most of the coastal zone of the Mediterranean Sea in the 1990s and 2000s, with a negative impact on both biodiversity and structure in the invaded ecosystems.

Within a mixed meadow of *C. nodosa* and *Nanozostera noltei* (Hornemann) Tomlinson & Posluszny (as *Zostera noltii* Hornemann) in the Tyrrhenian Sea, this invasive NIS considerably reduced the shoot density of the former [29] and altered the density of its flowering plants [30]. However, if the ecology of native species and NIS is significantly different, no interaction between the species can be established [28]. In Mar Piccolo, no NIS has caused serious damage to benthic populations to date [26]. Concerning seaweeds, the only species with an invasive behavior is *Hypnea corona* Huisman & Petrocelli, which has been spreading in the basin since 2000, becoming a dominant species in the summer, but without any evident disruption for either the macrophytobenthic communities or the native fauna [26]. Mixed communities with native species of seaweed and invertebrates, and even *C. nodosa*, are observed in the basin (Figure 5a). Among NIS invertebrates, seven molluscan species were recorded in Mar Piccolo. The grazer *Bursatella leachii* Blainville 1817 and the carnivorous *Melibe viridis* (Kelaart, 1858) are frequently present in the *C. nodosa* meadow, both in Mar Piccolo and Mar Grande of Taranto [31] (Figure 5b), as well as other coastal Mediterranean environments [32–36]. Both species surely find refuge among the dense shoots, as well as their preferred meal. The former is primarily drawn to the rich epiphytic communities (e.g., diatoms and cyanobacteria) and the detritus, but sometimes does not mind associated seaweeds [35]. The latter finds several prey, such as microalgae herbivores, filter feeders, and ectoparasites [12].

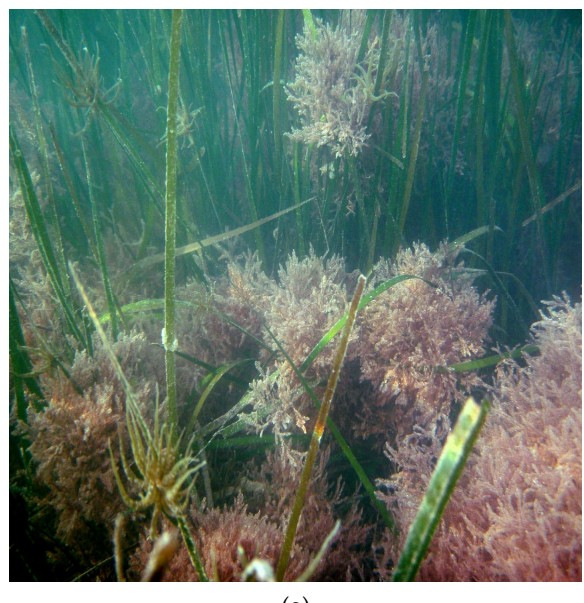
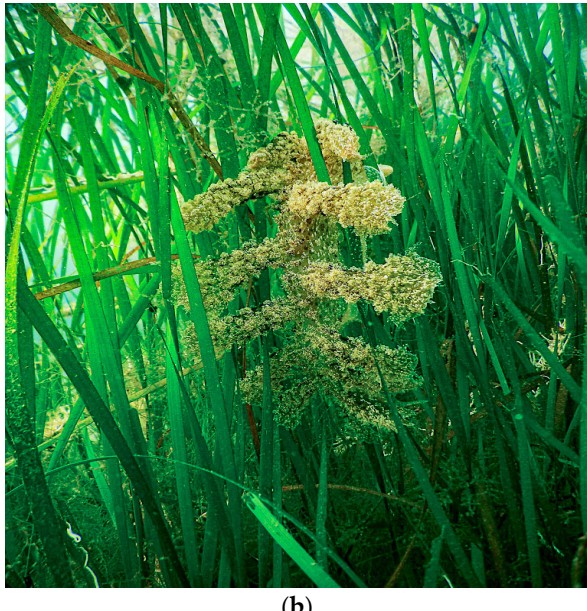

(**a**)　　　　　　　　　　　　　　　(**b**)

**Figure 5.** In the last few decades, *Cymodocea nodosa* shoots have become a good refuge for alien organisms in Mar Piccolo of Taranto: (**a**) Thalli of *Hypnea corona* intertwined with leaves (photo credit: Giuseppe Portacci); (**b**) A specimen of *Melibe viridis* hidden within the dense meadow.

Once again, the effectiveness of LTER activities in the study of biodiversity in Mar Piccolo of Taranto is made evident. The continuous monitoring and the availability of historical observations allow us to follow the population development of species worth protecting (e.g., *Pinna nobilis* and *Cymodocea nodosa*). The fate of the NIS and every possible new introduction are under continuous surveillance. New technologies, such as remote sensing and eDNA, combined with field activities and the classical taxonomy will surely lead to a more in-depth knowledge. In Italy, politicians have become more aware of this. The funding of the National Biodiversity Future Center, since June 2022, which has as its main task "*conserve, restore, monitor and enhance Italian and Mediterranean biodiversity*", is a step towards full recognition of the importance of biodiversity studies.

**Author Contributions:** Conceptualization, G.D. and A.P.; methodology, G.D and F.R.; investigation, G.D., A.P., and F.R.; resources, G.D. and F.R.; data curation, A.P.; writing—original draft preparation, A.P.; writing—review and editing, E.C., G.D., A.P., and F.R.; visualization, A.P.; supervision, A.P. and F.R.; project administration, A.P.; funding acquisition, A.P. and F.R. All authors have read and agreed to the published version of the manuscript.

**Funding:** This research was funded by POR PUGLIA FESR-FSE 2014/2020—Axis VI, Action 6.5.; under the National Recovery and Resilience Plan (NRRP), Mission 4 Component 2 Investment 1.4—Call for tender No. 3138 of 16 December 2021, rectified by Decree n. 3175 of 18 December 2021 of the Italian Ministry of University and Research funded by the European Union—NextGenerationEU, Project code CN_00000033, Concession Decree No. 1034 of 17 June 2022 adopted by the Italian Ministry of University and Research, CUP B83C22002930006 Project title "National Biodiversity Future Center—NBFC"; and under the NRRP, Mission 4 Component 2 Investment 3.1, code IR_0000032, CUP B53C22002150006 Project title "ITINERIS". The APC was funded by POR PUGLIA FESR-FSE 2014/2020—Axis VI, Action 6.5.

**Acknowledgments:** Giovanni Squitieri (www.officinadellimmagine.eu) is acknowledged for his participation in all of the underwater activities and for the photo documentation produced. We also acknowledge the contribution of Giuseppe Portacci for sampling activities and the photo provided.

**Conflicts of Interest:** The authors declare no conflicts of interest. The funders had no role in the design of the study; in the collection, analyses, or interpretation of data; in the writing of the manuscript; or in the decision to publish the results.

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
