# Peer review of "Native and Non-Indigenous Biota Associated with the Cymodocea nodosa (Tracheophyta, Alismatales) Meadow in the Seas of Taranto (Southern Italy, Mediterranean Sea)"

_diversity, doi:10.3390/d16070368_

Round 1
Reviewer 1 Report
Comments and Suggestions for Authors
The authors have described a improvement of a Cymodocea nodosa in a coastal system in the southern Italy, based on a long term ecological research. This is an interesting showcase for the need to support these long-term iniciatives. There are only a few corrections that I would suggest.
Comments on the Quality of English LanguageLine 41: Instead of "diving observations" use "Underwater observations by scuba divers"
Line 52: "(...) a sign of a high <ecological> status(...)"
Line 56: The abbreviation NRRP appears for the first time in this line, please provide the meaning here, instead of in line 61.
Line 62: I would suggest to use "underwater observations" instead of "diving observations"
Line 81: "an infection by the (...)"
Line 84: Write "like most transitional water systems" instead of "like most of the transitional water systems"
Line 125: "In Italy, finally there was an awareness of this". Please rephrase... who is aware? Since when? It is a nice final remark for the manuscript, so you can improve the sentence a bit.
Author Response
Dear Referee 1
Thank you very much for taking the time to review this manuscript. Please find the detailed responses below and the corresponding revisions highlighted in red in the re-submitted file
Comments 1: Line 41: Instead of "diving observations" use "Underwater observations by scuba divers"
Response 1: Thank you for pointing this out. We agree with this comment. Therefore, we have changed "diving observations" in "Underwater observations by scuba divers", as suggested, at Page 1, line 41
Comments 2: Line 52: "(...) a sign of a high <ecological> status(...)"
Response 2: Thank you for pointing this out. We agree with this comment. Therefore, we have added "ecological" as suggested at Page 2, line 52
Comments 3: Line 56: The abbreviation NRRP appears for the first time in this line, please provide the meaning here, instead of in line 61.
Response 3: Thank you for pointing this out. We agree with this comment. Therefore, we have added "National Recovery and Resilience Plan" at page 2, line 56, and deleted at line 61
Comments 4: Line 62: I would suggest to use "underwater observations" instead of "diving observations"
Response 4: Thank you for pointing this out. We agree with this comment. Therefore, we have changed "diving observations" in "underwater observations", as suggested at page 2, line 62
Comments 5: Line 81: "an infection by the (...)"
Response 5: Thank you for pointing this out. We agree with this comment. Therefore, we have changed "of" in "by" as suggested, at page 3 line 81
Comments 6: Line 84: Write "like most transitional water systems" instead of "like most of the transitional water systems"
Response 6: Thank you for pointing this out. We agree with this comment. Therefore, we have changed "most of the" in "most" as suggested, at page 3 line 84
Comments 7: Line 125: "In Italy, finally there was an awareness of this". Please rephrase... who is aware? Since when? It is a nice final remark for the manuscript, so you can improve the sentence a bit.
Response 7: Thank you for pointing this out. We agree with this comment. Therefore, we have changed the phrase, changing "there was an awareness" in "also politics became aware" at page 5, line 125.
At line 126, "since June 2022" was added.
The final statement "can be an initial step and the fulfillment of its main task “conserve, restore, monitor and enhance Italian and Mediterranean biodiversity” is on track" was changed to "which has as its main task “conserve, restore, monitor and enhance Italian and Mediterranean biodiversity”, can be an initial step towards the full recognition of the importance of biodiversity studies".
Reviewer 2 Report
Comments and Suggestions for Authors
This MS is a so-called "Interesting images from the sea" and as such is full-filling most of the requirements described. The MS gives an insight to the current condition of the Cymodocea nodosa seagrass in Southern Italy. Here the euthophication has decreased and the Cymodocea recovers successfully providing a habitat to a variety of both invasive and non-invasive species illustrated by high quality photos. As I understand it these Cymodocea meadows are not affected by the very invasive species Caulerpa disturbing the ecosystems elsewhere in the Mediterranean. Contrary, Cymodocea offers other invasive species a "warm welcome" as indicated in the title. I only have a few comments to the MS. In the title it should be indicated that the "warm welcome" is towards new species. Also in the instructions to authors it is mentioned that the fig. legend should be long and detailed and contain references. This is not full-filled in the MS. I miss a map of the locations photographed. I recommend acceptance after minor revision.
Author Response
Dear Referee 2
Thank you very much for taking the time to review this manuscript. Please find the detailed responses below and the corresponding revisions highlighted in red in the re-submitted file
Comments 1: In the title it should be indicated that the "warm welcome" is towards new species
Response 1: Thank you for pointing this out. We agree with this comment, but we have changed the title in "Native and non-indigenous biota associated with the Cymodocea nodosa (Tracheophyta, Alismatales) meadow in the seas of Taranto (southern Italy, Mediterranean Sea)", following the Editor suggestion
Comments 2: Also in the instructions to authors it is mentioned that the fig. legend should be long and detailed and contain references. This is not fullfilled in the MS
Response 2: Thank you for pointing this out. We agree with this comment, we have changed the legends, taking as an example the legends of one of the last papers published in this topical collection. Below you find the changes.
Figure 1. Associated fauna to Cymodocea nodosa shoots in the Mar Grande of Taranto:
becomes
Figure 2. In the Mar Grande of Taranto, a Cymodocea nodosa meadow flourishes along the south-eastern coast, and hosts a diverse associated fauna among shoots:
Figure 2. A dense meadow of Cymodocea nodosa
becomes
Figure 3. Starting from 2000, due to the bettering of the environmental conditions, a dense meadow of Cymodocea nodosa developed
Figure 3. Pinna nobilis among the leaves of Cymodocea nodosa in the Mar Piccolo of Taranto
becomes
Figure 4. Until 2018, it was common to observe healthy specimens of Pinna nobilis among the leaves of Cymodocea nodosa in the Mar Piccolo of Taranto
Figure 3. Alien organisms among Cymodocea nodosa shoots
becomes
Figure 5. In the last decades, Cymodocea nodosa shoots became a good refuge also for alien organisms among Cymodocea nodosa shoots
Comments 3: I miss a map of the locations photographed
Response 3: Thank you for pointing this out. We agree with this comment, we added Figure 1 with the map